# Metabolic-Dysfunction-Associated Fatty Liver Disease and Gut Microbiota: From Fatty Liver to Dysmetabolic Syndrome

**DOI:** 10.3390/medicina59030594

**Published:** 2023-03-17

**Authors:** Ludovico Abenavoli, Giuseppe Guido Maria Scarlata, Emidio Scarpellini, Luigi Boccuto, Rocco Spagnuolo, Bruno Tilocca, Paola Roncada, Francesco Luzza

**Affiliations:** 1Department of Health Sciences, University “Magna Graecia”, 88100 Catanzaro, Italy; 2Translationeel Onderzoek van Gastro-enterologische Aandoeningen (T.A.R.G.I.D.), Gasthuisberg University Hospital, KU Leuven, Herestraat 49, 3000 Leuven, Belgium; 3School of Nursing, Healthcare Genetics Program, Clemson University, Clemson, SC 29634, USA; 4School of Health Research, Clemson University, Clemson, SC 29634, USA

**Keywords:** fatty liver, gut microbiota, dysbiosis, obesity, type 2 diabetes mellitus

## Abstract

Metabolic-dysfunction-associated fatty liver disease (MAFLD) is the recent nomenclature designation that associates the condition of non-alcoholic fatty liver disease (NAFLD) with metabolic dysfunction. Its diagnosis has been debated in the recent period and is generally associated with a diagnosis of steatosis and at least one pathologic condition among overweight/obesity, type 2 diabetes mellitus, and metabolic dysregulation. Its pathogenesis is defined by a “multiple-hit” model and is associated with alteration or dysbiosis of the gut microbiota. The pathogenic role of dysbiosis of the gut microbiota has been investigated in many diseases, including obesity, type 2 diabetes mellitus, and NAFLD. However, only a few works correlate it with MAFLD, although common pathogenetic links to these diseases are suspected. This review underlines the most recurrent changes in the gut microbiota of patients with MAFLD, while also evidencing possible pathogenetic links.

## 1. Introduction

Metabolic dysfunction-associated fatty liver disease (MAFLD) is a recent nomenclature that associates non-alcoholic fatty liver disease (NAFLD) with a condition of systemic metabolic dysfunction [1]. The diagnosis of MAFLD follows specific criteria, such as the detection of hepatic steatosis (with a diagnosis conducted by imaging, biomarkers, or histology) and at least one characteristic among overweight/obesity, type 2 diabetes mellitus (T2DM), and metabolic dysregulation. The last criterion requires the presence of at least two characteristics, including increased waist circumference, hypertension, hypertriglyceridemia, low high-density lipoprotein cholesterol (HDL-C), pre-diabetes, insulin resistance, and subclinical inflammation [2]. Lin and colleagues [3] were the first to compare the diagnostic criteria of MAFLD and NAFLD in 13,083 cases identified from the NHANES III database. Their data showed that the MAFLD population had higher liver enzymes and more glucose and lipid-metabolism-related disorders. According to the authors, this new definition of MAFLD can specifically identify more patients at risk of developing cirrhosis and liver cancer, as they are affected by metabolic syndrome [3]. Further investigations corroborated the important role of this new MAFLD definition, as it better identified patients with significant hepatic fibrosis (93.9% MAFLD vs. 73.0% NAFLD) [4] or high liver stiffness (adjusted beta 0.116, *p* < 0.001 MAFLD vs. adjusted beta 0.006, *p* = 0.90 NAFLD), with respect to the NAFLD criteria [5]. Furthermore, in 2306 subjects with fatty liver, MAFLD (Hazard Ratio; HR 1.08, 95% CI 1.02–1.15, *p* = 0.014) and alcohol consumption (20–39 g/day; HR 1.73, 95% CI 1.26–2.36, *p* = 0.001) were independently associated with the worsening of the Suita score to predict the progression of atherosclerotic cardiovascular risk vs. the NAFLD group (HR 0.70, 95% CI 0.50–0.98, *p* = 0.042) [6]. This important evidence led to the acceptance of the definition of MAFLD nomenclature in 2022 by the “Global multi-stakeholder consensus on the redefinition of fatty liver disease”, which includes more than 1000 signatories with different expertise backgrounds, from over 134 different countries [7]. From an epidemiological point of view, the prevalence of MAFLD varies between 26% and 39% [8,9,10,11] in the general population, up to 42% in inflammatory bowel diseases (IBD) [11], and 50.7% in overweight/obese adults [12]. This great heterogeneity relates to the use of different diagnostic techniques applied to various populations under investigation [11]. Finally, an imbalance of the gut microbiota composition, defined as dysbiosis, is involved in the development of MAFLD [13]. The present review underlines the most recurrent changes in the gut microbiota of patients with MAFLD, and hypothesizes their possible pathogenic links.

## 2. Materials and Methods

### Literature Review

A review of the literature was performed through PubMed, NCBI and Scopus search engines. Mesh terms were the keywords: “MAFLD”, “gut microbiota”, “probiotic”, “prebiotic”, “postbiotic”, “diet”, and “dysbiosis”. The search included English papers published in each period. All types of papers were included, i.e., reviews, retrospective analyses, and experimental studies. Figure 1 details a PRISMA flow diagram summarizing the short-listing procedure and reasons for exclusion of articles.

## 3. Gut Microbiota Changes in MAFLD Patients

Gut microbiota is an ecosystem composed of over 35,000 bacterial species, and performs several functions, such as nutrient and drug metabolism and antimicrobial protection, and it is involved in immunomodulation and the integrity of the gut barrier [14]. *Firmicutes* and *Bacteroidetes* constitute 90% of the microbial *Phyla* that characterize this ecosystem, while the remaining 10% are *Actinobacteria*, *Proteobacteria*, *Fusobacteria*, and *Verrucomicrobia* [15]. Specifically, the *phylum Firmicutes* is further characterized by more than 200 different genera, such as *Lactobacillus*, *Bacillus*, *Clostridium* (with a 95% abundance), *Enterococcus*, and *Ruminicoccus* [16]. On the other hand, the *Phylum Bacteroidetes* is characterized by fewer genera, such as *Bacteroides*, *Parabacteroides*, *Prevotella*, *Porphyromonas*, and *Alistipes* [17]. The two main phyla are in a delicate balance (measured as *Firmicutes/Bacteroidetes* ratio) with each other, thus maintaining the proper homeostasis of the gastrointestinal tract. This balance changes at various stages of life, from 0.4 in infants to 10.9 in adults and 0.6 in the elderly [18], and in dysmetabolic conditions, such as T2DM [19] and obesity [20]. For this reason, the *Firmicutes*/*Bacteroidetes* ratio is being studied as a biomarker of gut dysbiosis [21]. As previously reported, a condition of dysbiosis is involved in the development of MAFLD [13]. Few studies have analyzed the composition of the gut microbiota in MAFLD patients and healthy control subjects. In a retrospective cross-sectional study conducted by Zhang and colleagues, the authors analyzed the gut microbiota of 17 MAFLD patients with liver stiffness (LSM) ≥ 7.4 kPa (case group) and 68 control subjects with an LSM < 7.4 kPa (control group) [22]. Whole-genome sequencing from stool samples showed that *Bacteroidetes*, *Firmicutes*, and *Actinobacteria* were the most dominant *phyla* in the case group. At the genus and species level, *Prevotella copri*, *Phascolarctobacterium succinatutens*, *Eubacterium biforme*, and *Collinsella aerofaciens* were all more abundant in the case group than in the control group (*p* < 0.05). Furthermore, *Bacteroides coprocola* and *Bacteroides stercoris* were all reduced in the case group (*p* < 0.05). However, there was no statistically significant difference between the case and control groups (*p* > 0.05) using two different diversity indexes (α and β diversity, respectively). Regarding the correlation between dysbiosis and LSM, the levels of *Phascolarctobacterium succinatutens*, *Eubacterium biforme*, and *Collinsella aerofaciens* were also positively correlated with LSM (*p* < 0.05), while the levels of *Bacteroides stercoris* were inversely correlated with LSM (*p* < 0.05).

In another case-control study conducted by Yang and colleagues, the gut microbiota of 32 MAFLD patients and 30 healthy controls was analyzed using 16S ribosomal RNA (16S rRNA) sequencing from stool samples [23]. At the *phylum* level, the relative abundance of *Bacteroidetes*, *Proteobacteria*, and *Fusobacteria* increased, and *Firmicutes* decreased in MAFLD patients as compared to healthy control subjects. At the genus level, the relative abundances of *Prevotella*, *Bacteroides*, *Escherichia shigella*, *Megamonas*, *Fusobacterium*, and *Lachnoclostridium* increased, while *Clostridium*, *Agathobacter*, *Romboutsia*, *Faecalibacterium*, and *Blautia* decreased in MAFLD patients vs. healthy control subjects. Furthermore, the metabolomic analysis from stool and sera samples showed a reduction of hypoxanthine, propionyl carnitine, tyrosyl-alanine, hesperetin, methionine, and neohesperidin. As reported by the authors, flavonoids, such as hesperetin and neohesperidine, can reduce inflammatory cell infiltrations, hepatic steatosis and fibrosis, body weight, and insulin resistance in mice models. Finally, the authors set the inter-individual variability in the composition of the gut microbiota that is influenced by diet as a limitation of the study. The subjects enrolled did not follow the same diet before undergoing this study, so more standardized studies in patient selection are needed.

In a single-center prospective study conducted by Oh et al., the gut microbiota of 66 MAFLD patients and healthy controls was analyzed using 16S rRNA sequencing from stool samples [24]. The two groups showed different compositions of the gut microbiota. A statistically significant decrease of *Firmicutes* was observed in patients with MAFLD (50.08% in the MAFLD group and 60.15% in the healthy group; *p* < 0.001). *Proteobacteria* (10.69% vs. 3.09%; *p* < 0.001) and *Actinobacteria* (7.68% vs. 2.54%; *p* < 0.001) were significantly increased in patients with MAFLD compared to those in healthy control subjects. Finally, α-diversity showed statistically significant differences between the two groups (*p* < 0.001). In addition, a reduction in bacterial diversity for butyrate-producing microorganisms, such as *Anaerostipes*, *Coprococcus*, *Eubacterium*, *Roseburia*, *Faecalibacterium*, *Odoribacter*, *Oscillibacter*, *Subdoligranulum*, *Butyricimonas*, *Alistipes*, *Pseudoflavonifractor*, *Clostridium*, *Butyricicoccus*, and *Flavonifractor* was also shown in MAFLD patients. A reduction in the abundance of butyrate-producing bacteria implies an increase in intestinal permeability, with possible translocation of microorganisms to the liver, promoting the onset of MAFLD. The main limitation of the study includes not assessing the physical activity and lifestyle of the few subjects enrolled, as these factors play a key role in the composition of the gut microbiota and certainly need to be evaluated in further studies.

Yang et al. compared the gut microbiota of 20 patients with MAFLD, 20 patients with MAFLD and T2DM, and 19 healthy control subjects, using 16S rRNA sequencing from stool samples [25]. *Bacteroidetes*, *Firmicutes*, and *Proteobacteria* were the most abundant *phyla* in case groups (MAFLD and MAFLD + T2DM, respectively) vs. healthy control subjects (*p* < 0.05). This study showed significant differences among the groups regarding *Prevotellaceae*, *Cyanobacteria*, *Ruminococcaceae*, *Oscillospirales*, and *Clostridia* genera (*p* < 0.05). Finally, α and β diversity index analyses showed moderate differences in species among the groups under investigation. As previously reported, the diversity in the composition of the gut microbiota in the case population compared with the control population induces increased intestinal permeability, resulting in liver damage. Overall, although the cohort under review is rather small, this is one of the few case-control studies correlating T2DM and MAFLD. Indeed, according to the authors, assessment of the composition of the gut microbiota and its metabolites could be a reliable biomarker in the future.

Dorofeyev A. et al. analyzed the gut microbiota of 111 patients and 30 healthy control subjects, using 16S rRNA sequencing from stool samples [26]. The main group included 56 MAFLD + T2DM patients, the first group included 28 patients with MAFLD without T2DM and the second group included 27 patients with T2DM without MAFLD. The main group of patients, compared to healthy control subjects, showed a significant increase in levels of *Actinobacteria* (28.6% vs. 14.1%; *p* < 0.05), a decrease in *Bacteroidetes* (13.7% vs. 41.7%; *p* < 0.05) and an increase in the ratio of *Firmicutes/Bacteroidetes* (3.16% vs. 0.88%; *p* < 0.05). Significantly higher levels of *Actinobacteria* were found in the main group, as compared to the first group (28.6% vs. 19.8%; *p* < 0.05). When these groups were compared against the second group, the data revealed higher levels of *Actinobacteria* (28.6% vs. 17.1%; *p* < 0.05), lower levels of *Bacteroidetes* (13.7% vs. 32.4%; *p* < 0.05) and an increased *Firmicutes/Bacteroidetes* ratio (3.16% vs. 1.06%; *p* < 0.05). The comparison between the first group and healthy controls showed significantly lower levels of *Bacteroidetes* (21.1% vs. 41.7%; *p* < 0.05) and an increase in the ratio of *Firmicutes/Bacteroidetes* (2.26% vs. 0.88%; *p* < 0.05). Regarding the second group, the authors found only a significant increase in “other” microorganisms compared to the control group (15.8% vs. 6.9%; *p* < 0.05). According to the authors, the composition of the gut microbiota is strongly influenced by the population under investigation. Indeed, these changes in the composition of the gut microbiota in cases from Ukraine, compared to controls, could be associated with genetic characteristics, dietary habits, and the use of hypoglycemic drugs. A schematic representation of gut microbiota dysbiosis in MAFLD patients and MAFLD patients with T2DM is shown in Figure 2.

## 4. MAFLD and Gut Microbiota: Possible Pathogenetic Ways

Although the pathogenetic link between gut microbiota and NAFLD has been widely investigated [27,28], its relationship with MAFLD is poorly known. It is well known that MAFLD is a “multiple-hit” disease, which has obesity, diabetes, insulin resistance, genetic and environmental factors, and a dysbiosis of the gut microbiota as risk factors [29], as shown in Figure 3.

The human gut, after being colonized by microorganisms, manages to maintain a state of homeostasis due to continuous regulation by the immune system [30]. In addition, the diet can facilitate this delicate balance, promoting the integrity of the mucosal barrier and disfavoring the translocation of intestinal pathogens [31]. The structure of the intestinal barrier is maintained strong by tight junctions consisting of transmembrane single-span (such as junctional adhesion molecules) or tetraspan proteins (such as occludin, claudin, and tricellulin) [32]. Additional proteins that serve as scaffolds are *zonula occludens* proteins [33]. The junctional adhesion molecule A (JAM-A) was studied by Rahman et al. in mouse models, with interruption of the *F11r* gene encoding for JAM-A. Male C57BL/6 (control) or *F11r^−/−^* mice were fed differently for 8 weeks: first a normal diet, then a diet with a high content of saturated fat, fructose, and cholesterol. The diet rich in saturated fat, fructose, and cholesterol increased the abundance of *Proteobacteria* and *Firmicutes* and reduced the abundance of *Bacteroidetes* in the lumen of control and *F11r^−/−^* mice. Furthermore, after being fed a diet rich in saturated fat, fructose, and cholesterol, the *F11r^−/−^* mice showed histological evidence of severe steatosis and lobular inflammation, in contrast to the moderate steatosis developed by control mice that had been fed the same diet. In addition, decreased expression of JAM-A was correlated with increased mucosal inflammation. This event, according to the authors, is related to a compensatory mechanism in the maintenance of the intestinal barrier function in the absence of JAM-A, corroborated by the increased expression of occludin and claudin-4 in the colon of *F11r^−/−^* mice fed a normal diet, as compared to control mice [34]. Clinical trials showed that the reduced levels of bacteria of the genus *Akkermansia* found in MAFLD patients compared to healthy controls are the consequence of a reduction in fermentation products, including butyrate and acetate. These short-chain fatty acids are critical in maintaining the homeostasis of the microbiota and in the structure of the gut [35,36]. Overall, the role of *Akkermasia* in obesity and metabolic disorders warrants further investigation using clinical models. In fact, in pre-clinical models with an abundance of this bacterium, prebiotics, and polyphenols have been shown to have positive effects on metabolic disorders [37]. Similar events result in increased intestinal permeability, known as “leaky gut” [38], with a transition of lipopolysaccharide (LPS), a component of the Gram-negative outer wall, to the liver via the portal vein [39]. The interaction between LPS and Toll-like receptor 4 (TLR4) expressed by Kuppfer cells, activates the nuclear factor kappa-light-chain enhancer of activated B cells (NF-κB) and subsequently an inflammatory cascade, which is a pathogenetic mechanism shared with NAFLD [40].

### 4.1. Gut Microbiota Dysbiosis and Obesity

Dysbiosis of the gut microbiota is related to other pathogenic factors that contribute to the development of MAFLD. It is known how the gut microbiota regulates certain mechanisms that lead to obesity [41]. The gut microbiota ferments carbohydrates into short-chain fatty acids (SCFAs), which, after intestinal absorption, promote energy homeostasis [42]. In the enterocyte, the three main SCFAs, acetate, butyrate, and propionate, are converted to acetyl-CoA by acetyl-CoA carboxylase to produce adenosine triphosphate (ATP) through the Krebs cycle. This pathway contributes to maintaining cellular homeostasis, consequently strengthening tight-junction function and intestinal barrier integrity [43]. Despite this “beneficial” role, a dysbiosis of the gut microbiota can lead to increased SCFA production, resulting in lipid accumulation in the liver [44]. This mechanism could support a possible pathogenetic link between gut dysbiosis, obesity, and MAFLD. In addition, microorganisms in the gastrointestinal tract are involved in the production of hormones that positively or negatively regulate satiety, such as leptin, insulin, and ghrelin [45]. Specifically, neurons that express pro-opiomelanocortin (POMC) and cocaine- and amphetamine-regulated transcript (CART) in the hypothalamic arcuate nucleus are the targets of leptin binding to them to inhibit hunger signals [46]. Ghrelin is a 28-amino-acid peptide present in our organism in two different iso-forms: n-octanoyl modified ghrelin, and des-acyl ghrelin [47]. The acylated form is mainly involved in the orexigenic role by stimulating the synthesis of neuropeptide Y (NPY) and agouti-related protein (AgRP) in neurons in the arcuate nucleus of the hypothalamus and cerebellum, which promote increased food intake [48]. Insulin-like peptide 5 (Insl5) is a two-chains peptide hormone member of the relaxin family of peptides, with a structure similar to insulin [49]. Its biochemical pathway involves the G-protein-coupled relaxin/insulin-like family peptide receptor 4 (Rxfp4) to carry out its orexigenic action [50]. In pre-clinical models, Insl5 increased food intake in wild-type mice with respect to mice models without the Rxfp4 receptor. Furthermore, plasmatic Insl5 levels were increased in fasting, but decreased with feeding [51]. Although further studies in clinical and pre-clinical models are needed, the regulation of these hormones could be used to improve body weight or metabolism [52]. Table 1 summarizes case-control studies regarding metabolite production and obesity.

Overall, studies have focused on characterizing the diversity in the composition of the gut microbiota non-lean NAFLD, lean NAFLD, and healthy control subjects. A recent review showed the following differences in gut microbiota composition: (i) both NAFLD groups had a decrease in *Firmicutes* and *Ruminococcaceae*, but a decrease in *Leuconostocaceae* was only observed in obese NAFLD; (ii) an increase in the *Bacteroidetes/Firmicutes* ratio in lean and obese NAFLD patients compared to healthy control subjects; (iii) lean NAFLD patients showed an increase in *Ruminococcaceae* compared to obese NAFLD, and an increase in *Dorea* and a decrease in *Marvinbryantia* and *Christensellenaceae* compared to healthy control subjects [58]. However, there is a lack of case-control studies regarding MAFLD, obesity, and gut microbiota composition.

### 4.2. Gut Microbiota Dysbiosis, T2MD, and Insulin Resistance

Another support for the pathogenesis of MAFLD may be the close correlation between gut microbiota, T2DM, and insulin resistance. T2DM is associated with an over-production of pro-inflammatory cytokines, such as interleukin (IL)-1α, IL-6, IL-10, and IL-22 [59]. The role of the gut microbiota is to modulate the inflammatory response by secreting anti-inflammatory cytokines. For example, *Roseburia intestinalis* promotes the production of IL-22, a cytokine with anti-inflammatory action, while reducing insulin resistance and diabetes initiation [60]. *Bacteroides fragilis* polysaccharide A induces IL-10 secretion by B and T cells obtaining the reduced inflammatory process in the gut [61]. In knockout mouse models of the IL-6 gene, the absence of this cytokine led to significantly increased expression of defensins α3 and α4 in the gut, promoting microbiota remodeling and subsequent inflammatory response [62]. Similarly, in IL-1α-deficient mice, dysbiosis of the gut microbiota had been found, resulting in an inflammatory intestinal state vs. wild-type mouse models [63]. The gut microbiota is closely related to T2DM, as it can regulate insulin clearance [64]. Foley et al. found impaired insulin clearance after 6 weeks in mouse models lacking gut microbes and after a fat-rich obesogenic diet, compared with generally healthy mice treated with a control diet [65]. Additionally, insulin resistance is associated with increased intestinal permeability under conditions of dysbiosis, without necessarily being influenced by obesity [46]. Reduced adiponectin concentrations and increased leptin concentrations were associated (*p* < 0.05) with obesity, while Zonulin expression is positively associated (*p* < 0.05) with body mass index and insulin concentration. In addition, elevated insulin production was associated with increased intestinal barrier permeability [66]. Overall, dysbiosis of the gut microbiota that promotes T2DM and insulin resistance may lead to considering MAFLD a hepatic phenotype of systemic insulin resistance [67].

### 4.3. Gut Microbiota Dysbiosis and Genetic Factors

Lastly, genetic and environmental factors are continuously investigated regarding the predisposition and pathogenesis of MAFLD. Certainly, the mechanism by which specific single nucleotide polymorphisms (SNPs) are inherited plays a key role in susceptibility to the development of the disease. These include *Patatin-like Phospholipase Domain -containing 3* (*PNPLA3*) and *Membrane-Bound O-acyltransferase Domain-containing 7* (*MBOAT7*) [68]. Specifically, the rs641738 variant of the *MBOAT7* gene, which promotes the regulation of insulinemia, has been evaluated as a predisposing factor of hepatocellular carcinoma (HCC), even in the absence of cirrhosis in MAFLD patients [69]. A recent study conducted on 564 MAFLD patients and healthy control subjects showed that the CC genotype of the *PNPLA3* gene (encoding for a triacylglycerol lipase that mediates the hydrolysis of triacylglycerol in adipocytes) rs738409 and the TT genotype of the *MBOAT7* gene rs64173 are risk factors for the occurrence of MAFLD [70]. Another case-control study, conducted by Liao S. et al. on 286 MAFLD patients and 250 healthy control subjects, showed a correlation between the *PNPLA3* rs738409 variant and MAFLD (odds ratio [OR] = 1.791 and 1.377, respectively, *p* = 0.038 and 0.027, respectively) and with aspartate aminotransferase (AST) levels [71]. On the other hand, dysbiosis of the gut microbiota in association with genetic mutations is called “genetic dysbiosis” [72]. This is characterized by two possible events: (i) mutation of genes encoding pattern recognition receptors (PRRs) failing bacterial recognition [73]; (ii) mutations in genes involved in the regulation of the immune response, resulting in stimulation of pro-inflammatory cytokines and diffuse inflammation in the gut, affecting the composition of the microbiota [74]. Moreover, they are certainly related to the onset of IBD, which in turn is closely related to MAFLD [71]. Overall, genetic factors that promote dysbiosis of the gut microbiota may contribute to the pathogenesis of MAFLD.

### 4.4. Dysmetabolic Comorbidities and MAFLD Progression

The evidence cited so far underlines the pathogenic multifactorial nature of MAFLD, in which inter-individual factors and dysmetabolic comorbidities promote its onset and progression. Alteration in the composition of the gut microbiota is an important factor in its occurrence, through pathways that disfavor the production of metabolites by affecting the integrity of the gut barrier and increasing its permeability. The passage of microorganisms and Gram-negative LPS via the portal vein to the liver is carried out by an inflammatory process. In addition, the cascade mechanism of pro-inflammatory cytokines is already widely represented in patients with dysmetabolic diseases such as T2DM, insulin resistance, and obesity. Moreover, the overproduction of SCFA under dysbiosis conditions is responsible for the accumulation of liver fat in these patients. All these interconnected mechanisms could explain the etiopathogenesis of MAFLD, which still deserves further investigation.

## 5. Therapeutic Approaches

Possible therapeutic approaches are related to re-establishing the eubiosis condition of the gut microbiota, and thus the correct balance of the microbial community within [75]. Diet is an important factor that influences the composition of the gut microbiota, which is the reason why a balanced diet can promote its eubiosis [76]. The Mediterranean diet was purposed as a possible therapeutic approach [77]. However, studies aimed at investigating proper dietary intake are needed for better management of MAFLD patients [78]. We know more about the preventive role of the Mediterranean diet in NAFLD [79]. This dietary regimen is composed mainly of a higher intake of fish and vegetables than of meat and dairy products, which disadvantages the onset of many diseases, such as T2DM, obesity, and NAFLD [80]. The antioxidants (such as polyphenols) in this diet promote the reduction of the inflammatory state typical of these diseases [81,82], acting at different levels: (i) modulating the pathway of mitogen-activated protein kinases (MAPKs), resulting in reduced production of pro-inflammatory cytokines [83]; (ii) inhibiting NF-κB-induced pro-inflammatory gene expression at multiple levels [84]; and (iii) inhibiting cyclo-oxygenases (COX) with reduced prostaglandin synthesis [85]. This dietary approach—rich in vegetables and antioxidants—has “healthy” consequences on the remodeling of the gut microbiota by promoting the growth of good bacteria that promote SCFA synthesis and degrade toxic metabolites [86]. Overall, the prevention of these diseases may be related to the prevention of MAFLD progression [87].

As reported by The International Scientific Association for Probiotics and Prebiotics, probiotics are defined as “live microorganisms that, when administered in adequate amounts, confer a health benefit to the host” [88], while prebiotics are “a selectively fermented ingredient that results in specific changes in the composition and/or activity of the gastrointestinal microbiota, thus conferring benefit(s) upon host health” [89]. Subsequently, the same International Scientific Association defined postbiotics as “a preparation of inanimate microorganisms and/or their components that confers a health benefit on the host” [90]. While the preventive and therapeutic role of probiotics in NAFLD is quite clear in clinical [91,92] and pre-clinical models [93,94], further investigations are needed in patients with MAFLD. A recent meta-analysis showed that the use of probiotics holds promise for reducing liver enzyme levels in patients with MAFLD. Among a total of 772 patients, the use of probiotics for therapeutic purposes could reduce the levels of alanine aminotransferase (mean difference (MD): −11.76 (−16.06, −7.46), *p* < 0.00001), aspartate aminotransferase (MD: −9.08 (−13.60, −4.56), *p* < 0.0001), γ-glutamyltransferase (MD: −5.67 (−6.80, −4.54), *p* < 0.00001) and homeostasis model assessment of insulin resistance (MD: −0.62 (−1.08, −0.15), *p* = 0.01) in patients with MAFLD, compared to control patients. Indeed, this study did not show statistical significance for levels of total cholesterol, triglycerides, low-density lipoproteins (LDL), C-reactive protein (PCR), and tumor necrosis factor-α (TNF-α) [95]. Regarding prebiotics, their use, in combination with probiotics, is recommended, as suggested by a recent review that underlines the importance of this combined approach that showed a significant reduction in the levels of hepatic steatosis, alanine aminotransferase (ALT), AST, HDL, LDL, triglyceride and cholesterol levels in 782 MAFLD patients compared to healthy controls [96]. Finally, the use of postbiotics in mouse models promoted insulin sensitivity, whereas the use of ursodeoxycholic acid in human models showed a reduction in transaminases and insulin resistance [97]. Despite this, the literature about clinical trials that promote the use of postbiotics in MAFLD patients is still lacking. Overall, while the diagnostic approach regarding MAFLD has been clarified, international guidelines for the treatment of this disease are needed.

## 6. Conclusions and Future Directions

MAFLD is generally related to other disease states, such as T2DM, obesity, and insulin resistance, and constitutes a serious public health burden. A dysbiosis of the gut microbiota plays a key role in this context. Specifically, the increased permeability of the intestinal barrier, known as “leaky gut,” allows the passage of toxic products, such as LPS from Gram-negative bacteria, through the portal vein to the liver. However, the complex pathways involved in such dysbiosis deserve further investigation via the planning of new case-control studies. On the other hand, the preventive and therapeutic use of a diet rich in polyphenols, such as the Mediterranean diet, and the combined use of probiotics and prebiotics are widely recommended in the management of MAFLD patients. However, the literature on clinical trials related to these patients is still poor. For this reason, although the treatment of NAFLD patients is the subject of specific international guidelines, the therapeutic approach to be used with MAFLD patients is still under investigation. Expanding our knowledge of the active role that the gut microbiota can play in the pathogenesis of MAFLD could also facilitate the development of international guidelines for the prevention and treatment of this disease.

## Figures and Tables

**Figure 1 medicina-59-00594-f001:**
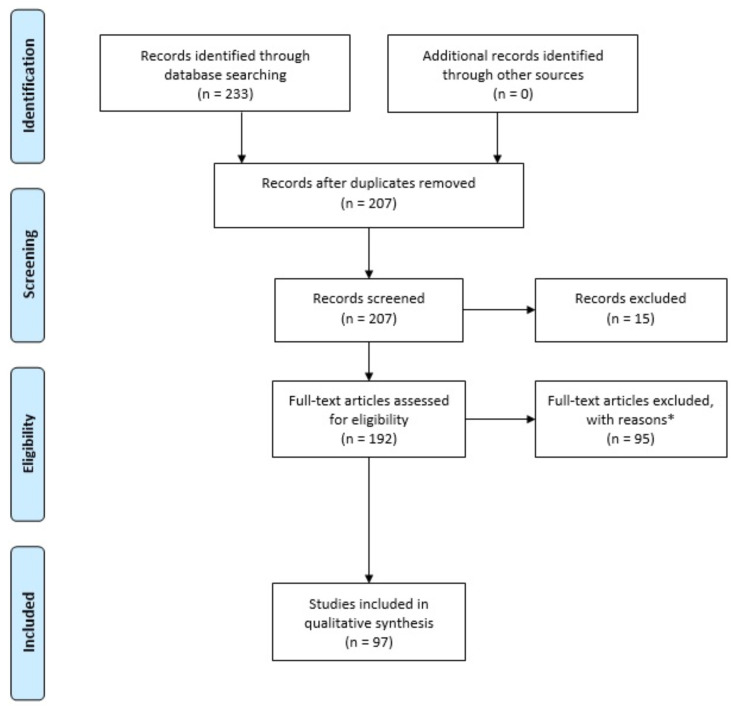
PRISMA flow diagram. * Full-text articles were excluded due to the following reasons: (1) the articles did not report data for individual comparison groups; (2) the articles did not distinguish NAFLD and MAFLD.

**Figure 2 medicina-59-00594-f002:**
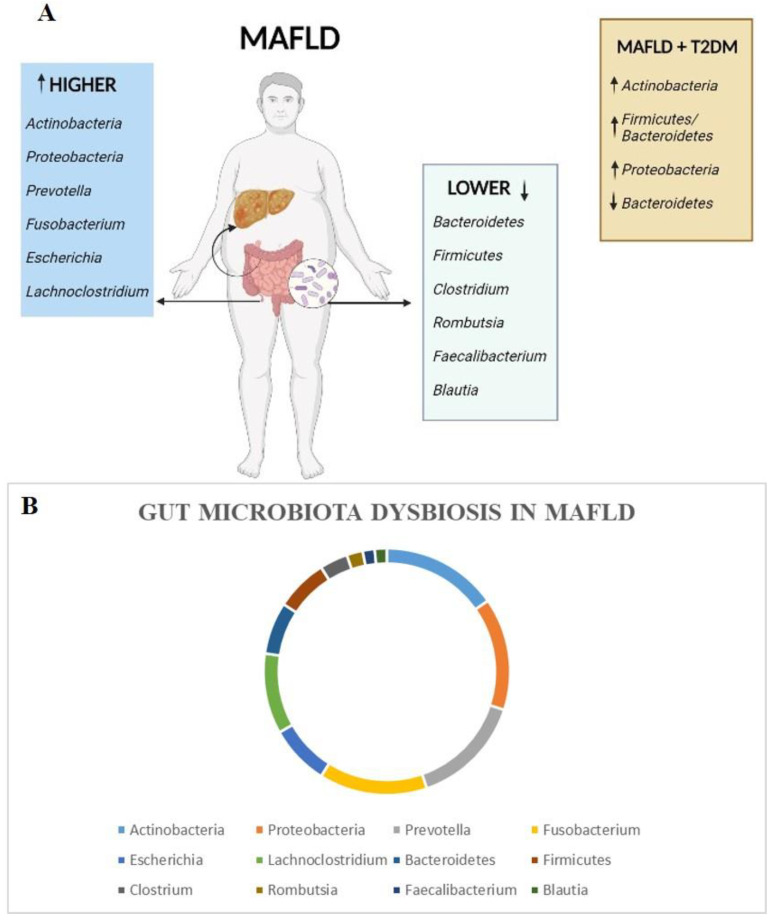
Schematic representation of gut microbiota dysbiosis in MAFLD patients (light blue and green boxes; panel (**A**,**B**)) and MAFLD + T2DM patients (orange box; panel (**A**)).

**Figure 3 medicina-59-00594-f003:**
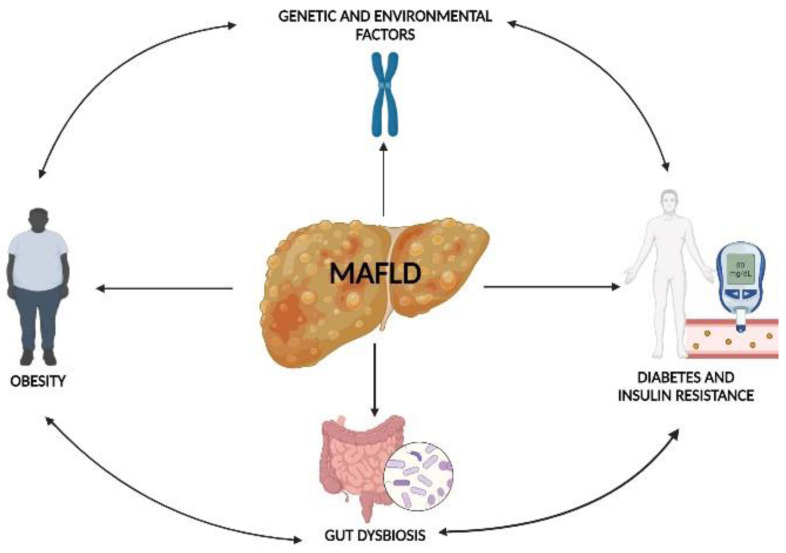
“Multiple-hit” hypothesis in MAFLD pathogenesis.

**Table 1 medicina-59-00594-t001:** Summary table of case-control studies regarding metabolite production and obesity.

Sample Size	Metabolites Involved	Biological Samples Analyzed	Results	References
208 obese subjects vs. 191 normal-weight subjects	Acetate, propionate, valerate, butyrate	Serum and stool	Higher concentrations of acetate (SMD = 0.87, 95% CI = 0.24–1.50), propionate (SMD = 0.86, 95% CI = 0.35–1.36), valerate (SMD = 0.32, 95% CI = 0.00–0.64) and butyrate (SMD = 0.78, 95% CI = 0.29–1.27) in obese subjects vs. normal-weight subjects	Kim KN et al., 2019 [53]
13 obese subjects vs. 13 normal-weight subjects	Acetate, butyrate	Stool	Acetate and butyrate were significantly higher in the group of obese patients compared to normal-weight patients (*p* = 0.033 and *p* = 0.004, respectively)	Martínez-Cuesta et al., 2021 [54]
92 obese adults vs. 92 normal-weight subjects	Leptin	Serum	Higher levels of leptin (51.24 ± 18.12 vs. 9.10 ± 2.99: *p* < 0.0001) in obese adults as compared to healthy control subjects	Kumar et al., 2020 [55]
35 obese adults vs. 20 normal-weight subjects	Leptin	Serum	Significant difference (*p* < 0.001) in leptin between the obese group (34.78 ± 13.96 ng/mL) and the non-obese control subjects (10.6 ± 4.2 ng/mL)	Al Maskari MY et al., 2006 [56]
1125 obese adults vs. 738 normal-weight subjects	Ghrelin	Serum	Lower levels of acyl ghrelin at baseline (SMD: −0.85; 95% CI: −1.13 to −0.57; *p* < 0.001) and postprandial at different time points (SMD 30 min: −0.85, 95% CI: −1.18 to −0.53, *p* < 0.001; SMD 60 min: −1.00, 95% CI: −1.37 to −0.63, *p* < 0.001; SMD 120 min: −1.21, 95% CI: −1.59 to −0.83, *p* < 0.001) in obese patients in respect to healthy control subjects	Wang Y. et al., 2022 [57]

## Data Availability

Not applicable.

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
