# Peer review of "Metabolic-Dysfunction-Associated Fatty Liver Disease and Gut Microbiota: From Fatty Liver to Dysmetabolic Syndrome"

_medicina, 2023, doi:10.3390/medicina59030594_

Round 1

Reviewer 1 Report

This is a very comprehensive paper on an emerging and important topic. I have enjoyed reading it. However, I do have several points that I would like to ask authors to address:

1. Methodology section could have been strengthened if the authors have followed PRISMA guidelines for literature selection. For example, recent review article on this topic have not been included. Please see the following: 10.15403/jgld-3308

2. The authors should have discussed in more details the differences in microbiota in lean vs non lean NAFLD/MAFLD patients. 

3. References are appropriate despite the fact that there are many from the first authors of the current study, I think this is reasonable given his research area. 

Author Response

This is a very comprehensive paper on an emerging and important topic. I have enjoyed reading it. However, I do have several points that I would like to ask authors to address:

  1. Methodology section could have been strengthened if the authors have followed PRISMA guidelines for literature selection. For example, recent review article on this topic have not been included. Please see the following: 10.15403/jgld-3308

REPLY 1: PRISMA guidelines for literature selection are now included in lines 66-71.

  1. The authors should have discussed in more details the differences in microbiota in lean vs non lean NAFLD/MAFLD patients.

REPLY 2: Differences in microbiota in lean vs non lean NAFLD/MAFLD patients were discussed in lines 256-265.

  1. References are appropriate despite the fact that there are many from the first authors of the current study, I think this is reasonable given his research area.

REPLY 3: Thank you for the comment. Our research group has been constantly working on these issues, hence our interest in writing this review.

Reviewer 2 Report

Metabolic dysfunction-Associated Fatty Liver Disease (MAFLD) is generally associated with a diagnosis of steatosis and at least one pathologic condition among over weight/obesity, type 2 diabetes mellitus, and metabolic dysregulation, whose pathogenesis is considered as alteration or dysbiosis of the gut microbiota. The authors review underlines the most recurrent changes in the gut microbiota of patients with MAFLD, facilitating the development of international guidelines for the prevention and treatment of this disease.

Most of this manuscript was well organized, some advice was as following:

1.     List out subtitles of the paragraphs, which would give precise idea to readers.

2.     Please present the data in lines 231-266 into a Table in the text.

Author Response

Metabolic dysfunction-Associated Fatty Liver Disease (MAFLD) is generally associated with a diagnosis of steatosis and at least one pathologic condition among over weight/obesity, type 2 diabetes mellitus, and metabolic dysregulation, whose pathogenesis is considered as alteration or dysbiosis of the gut microbiota. The authors review underlines the most recurrent changes in the gut microbiota of patients with MAFLD, facilitating the development of international guidelines for the prevention and treatment of this disease.

Most of this manuscript was well organized, some advice was as following:

  1. List out subtitles of the paragraphs, which would give precise idea to readers.

REPLY 1: Subtitles are now included in lines 223, 266, 292, 317.

  1. Please present the data in lines 231-266 into a Table in the text.

REPLY 2: Table 1 is now included in line 254-255.

Round 2

Reviewer 1 Report

The manuscript has been improved